# Neural Circuit Synthesis from Specification Patterns

**Frederik Schmitt**
CISPA Helmholtz Center for Information Security
Saarbrücken, Germany
`frederik.schmitt@cispa.de`

**Christopher Hahn**
CISPA Helmholtz Center for Information Security
Saarbrücken, Germany
`christopher.hahn@cispa.de`

**Markus N. Rabe**
Google Research
Mountain View, California, USA
`mrabe@google.com`

**Bernd Finkbeiner**
CISPA Helmholtz Center for Information Security
Saarbrücken, Germany
`finkbeiner@cispa.de`

## Abstract

We train hierarchical Transformers on the task of synthesizing hardware circuits directly out of high-level logical specifications in linear-time temporal logic (LTL). The LTL synthesis problem is a well-known algorithmic challenge with a long history and an annual competition is organized to track the improvement of algorithms and tooling over time. New approaches using machine learning might open a lot of possibilities in this area, but suffer from the lack of sufficient amounts of training data. In this paper, we consider a method to generate large amounts of additional training data, i.e., pairs of specifications and circuits implementing them. We ensure that this synthetic data is sufficiently close to human-written specifications by mining common patterns from the specifications used in the synthesis competitions. We show that hierarchical Transformers trained on this synthetic data solve a significant portion of problems from the synthesis competitions, and even out-of-distribution examples from a recent case study.

## 1 Introduction

In reactive synthesis, a circuit is automatically constructed from a logical specification given as a formula in linear-time temporal logic (LTL). LTL is widely used by the verification community and is the basis for industrial specification languages like the IEEE standard PSL [24]. Efficient synthesis tools for LTL would simplify the hardware design process: a hardware designer could focus on specifying *what* the circuit is supposed to compute, instead of implementing *how* the computation is done. LTL synthesis procedures, however, have to invoke involved reasoning engines, which turn often out to be infeasible when facing real-world problem instances. Much research has been conducted to push this form of hardware construction process closer to practice (see, for example, the synthesis of the AMBA protocol [5]). The high computational complexity of the general problem (2-EXPTIME-complete), however, is so far a barrier that seems insurmountable with classical, e.g., automaton-based, approaches. Recent successful applications of machine learning for logical tasks, such as SAT solving [45, 46], higher-order theorem proving [36, 3], and the LTL trace generation problem [22] encourage new approaches to the LTL synthesis problem using machine learning. Similar to the success of machine learning for program synthesis, e.g., [37, 20, 42], machine learning approaches might open a lot of possibilities in hardware synthesis. For example, secondary design

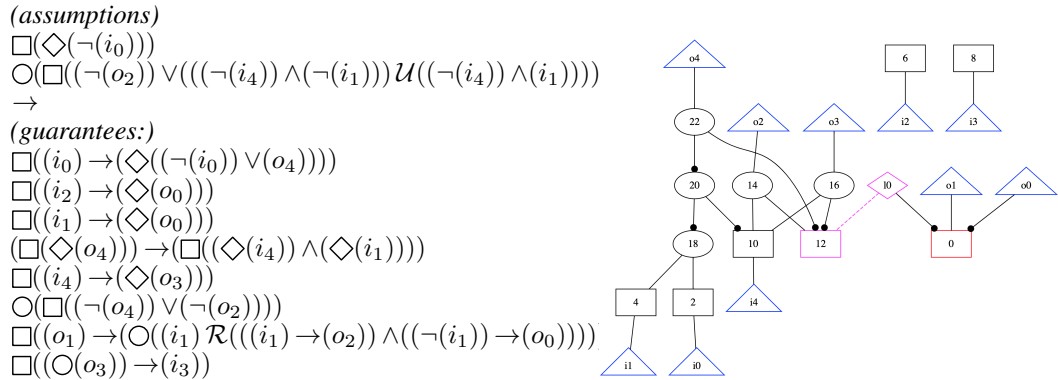

*(assumptions)*
$\Box(\Diamond(\neg(i_0)))$
$\bigcirc(\Box((\neg(o_2)) \vee (((\neg(i_4)) \wedge (\neg(i_1))) \, \mathcal{U} ((\neg(i_4)) \wedge (i_1)))))$
$\rightarrow$
*(guarantees:)*
$\Box((i_0) \rightarrow (\Diamond((\neg(i_0)) \vee (o_4))))$
$\Box((i_2) \rightarrow (\Diamond(o_0)))$
$\Box((i_1) \rightarrow (\Diamond(o_0)))$
$(\Box(\Diamond(o_4))) \rightarrow (\Box((\Diamond(i_4)) \wedge (\Diamond(i_1))))$
$\Box((i_4) \rightarrow (\Diamond(o_3)))$
$\bigcirc(\Box((\neg(o_4)) \vee (\neg(o_2))))$
$\Box((o_1) \rightarrow (\bigcirc((i_1) \, \mathcal{R} (((i_1) \rightarrow (o_2)) \wedge ((\neg(i_1)) \rightarrow (o_0))))))$
$\Box((\bigcirc(o_3)) \rightarrow (i_3))$

Figure 1: A specification in our test set, consisting of 2 assumption patterns and 8 guarantee patterns (left). A circuit, predicted by a hierarchical Transformer, satisfying the specification (right).

goals, which cannot be easily formalized, might be incorporated into the process using natural language. Applying machine learning to the area of hardware synthesis, however, suffers from a severe lack of sufficient amounts of training data.

In this paper, we consider a method to generate large amounts of additional training data, i.e., pairs of specifications and circuits implementing them. We show that hierarchical Transformers [33] can be trained on the circuit synthesis problem using the generated data and that the models can solve a significant portion of problems from the annual synthesis competition. In practice, logical hardware specifications follow specific design patterns [12]. To cope with the data scarcity of this problem, we propose a method that makes use of specification patterns, from which data for a successful training can be derived.

For example, a common LTL specification pattern looks as follows: $\Box(r \rightarrow \Diamond g)$. The formula describes a *response* property, stating that at every point in time ($\Box$), a request $r$ must be eventually ($\Diamond$) followed by a grant $g$. We obtain these patterns from the annual reactive synthesis competition [25]. We mined 2099 specification patterns from 346 benchmarks, which we split into assumption patterns and guarantee patterns. Assumption patterns restrict the space of possible inputs (environment behavior), and guarantee patterns describe how the circuit has to react to the environment. From these specification patterns, we generate larger specifications by conjoining assumption patterns to a specification $\varphi_A$ and by conjoining guarantee patterns to a specification $\varphi_G$. The implication $\varphi_A \rightarrow \varphi_G$ forms the final specification of the circuit. We obtained 200 000 specifications and used classical synthesis tools [14, 35] to compute circuits satisfying the specifications. Figure 1 shows an example held-out specification constructed in this fashion and a circuit predicted by one of our models (details on the data representation can be found in Section 3). When checking, the predicted circuit indeed satisfies the specification.

To train a machine learning model on the LTL synthesis task, we represent the decomposed specifications and circuits as sequences and use hierarchical Transformers [33]. We show that many of the model's predictions that differ from the circuits in our dataset satisfy the specifications when verifying the predictions[1], i.e., the model constructs a different, yet correct solution. When using a beam search, models achieve an accuracy of up to 79.9% on our synthetic test data and up to 66.8% on the original formulas from SYNTCOMP. The Transformer can even solve out-of-distribution formulas, taken from a recent case study [1], i.e., formulas that were not used for the specification pattern mining. Furthermore, the models can solve generated test instances on which classical LTL synthesis tools timed out. In practice, it is essential to handle both realizable (i.e., when a hardware implementation exists) and unrealizable (i.e., when no hardware implementation exists) specifications. We demonstrate that our approach achieves similar results on both realizable and unrealizable specifications.

---

[1]Note that verifying the solutions, i.e., model-checking, is a by-far easier problem (PSPACE vs 2-EXPTIME) and can typically be done in a fraction of the time needed to synthesize the circuits classically.

The remainder of this paper is structured as follows: Related work is presented in Section 2. The data representation and generation process is described in Section 3. The experimental setup and the experimental evaluation are presented in Section 4 and Section 5, respectively. We conclude the paper in Section 6.

## 2    Related Work

**Neural architectures for logical reasoning.**    Neural architectures for logical and mathematical reasoning have been studied recently. The closest work is the application of Transformers to the LTL trace generation problem demonstrating the generalization abilities of Transformers to the semantics of logics [22]. Despite the substantially greater complexity of the LTL synthesis problem, we are able to demonstrate the same generalization in this work. In addition, we consider both realizable and unrealizable specifications while for the LTL trace generation problem the satisfiability of LTL formulas was assumed. Lample and Charton trained Transformers on symbolic integration and solving differential equations and were able to outperform commercial systems on a synthetic dataset [31]. Similar to our findings Lample and Charton observed significant improvements in the Transformer's accuracy when using a beam search. Rabe et al. applied Transformers to formal mathematical statements and demonstrated the Transformer's reasoning abilities on tasks such as type inference and completing missing assumptions [41]. In contrast to the supervised setting in this work, Rabe et al. trained Transformers on an unsupervised skip-tree task that outperforms skip-sequence tasks for language modeling. For propositional logic Selsam et al. applied graph neural networks [43, 18] to solve the satisfiability problem [46]. In subsequent work Selsam and Bjørner applied the same architecture to the unsat-core prediction problem and demonstrated that their model can be used as a heuristic to speed up SAT solvers [45]. Lederman et al. applied graph neural networks to quantified Boolean formulas to learn heuristics for QBF solvers through deep reinforcement learning [32]. Paliwal et al. trained graph neural networks on higher-order logic terms to predict tactics for higher-order theorem proving [36]. When integrated with the DeepHOL [3] neural theorem prover the graph neural networks achieved state-of-the-art performance for higher-order proof search. Similar, Balunović et al. applied graph neural networks to SMT formulas to predict tactics for SMT solvers [2]. Strategies synthesized from their model demonstrated significant improvements over hand-crafted strategies from state-of-the-art SMT solvers. Earlier works on applying learning to mathematics, has focused on ranking premises or clauses Cairns [8], Urban [48, 49], Urban et al. [50], Meng and Paulson [34], Schulz [44], Kaliszyk and Urban [28].

**Classic synthesis tools.**    The hardware synthesis problem traces back to the definition of the problem by Alonzo Church in 1957 [11], thus also called Church's Problem. With theoretical solutions, already in 1969 by Büchi and Landweber [7], the field has matured today. From a foundational point of view, advances have been made algorithmically, e.g., with a quasi-polynomial algorithm for parity games [9], conceptually with distributed [40] and bounded synthesis [16], or expressiveness-wise, e.g., GR(1) [39] synthesis, which is an efficient fragment of LTL or synthesis for security properties [17]. From a practical point of view, the field can build on a rich supply of tools (e.g. [6, 15, 35]). The first synthesis competition (SYNTCOMP) [26] was held in 2014, as part of the annual international conference on computer-aided verification (CAV).

**Property specification patterns.**    Property specification patterns for temporal logics have already been identified by Dwyer et al. [12]. They proposed a general hierarchical specification pattern system containing 55 patterns that are mapped to formal specification languages such as LTL and CTL. More patterns for temporal logical formulas are identified by Etessami and Holzmann [13], Holeček et al. [23], Pelánek [38]. Konrad and Cheng [30] identified real-time specification patterns formulated in different real-time temporal logics and a structured English grammar. Grunske [19] presented a specification pattern system for probabilistic properties formulated in probabilistic temporal logic and a structured English grammar.

## 3    Datasets

In the following, we will first exemplary describe the specification language LTL and the circuit representation (the interested reader can find the full formalizations in the appendix). We will then describe our dataset, which is generated from specification patterns from the LTL track of SYNTCOMP 2020 [26].

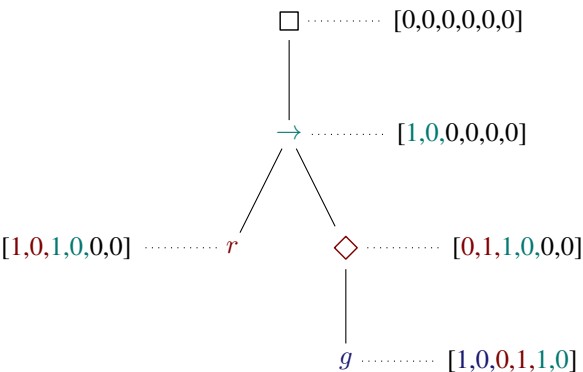

Figure 2: Example tree positional encoding for the LTL request pattern $\square(r \rightarrow \diamondsuit g)$.

## 3.1 LTL and And-Inverter Graphs

LTL can specify that some proposition $P$ must hold at every point in time ($\square P$) or that $P$ must hold at some future point of time ($\diamondsuit P$). By combining these operators, one can specify that $P$ must occur infinitely often ($\square \diamondsuit P$). The propositions are usually partitioned into inputs and outputs. In the following, we provide a small example. For inputs $r_1, r_2$ and outputs $g_1, g_2$ the LTL formula

$$
\begin{aligned}
& \square \neg(g_1 \wedge g_2) \\
\wedge\; & \square(r_1 \rightarrow \diamondsuit g_1) \\
\wedge\; & \square(r_2 \rightarrow \diamondsuit g_2)
\end{aligned}
$$

specifies a simple arbiter using a mutual exclusion property for grant $g_1$ and grant $g_2$ and two response properties that guarantee that always request $r_1$ is eventually answered by grant $g_1$ and always request $r_2$ is eventually answered by grant $g_2$. Given an LTL specification $\varphi$, i.e., an LTL formula $\varphi$ over atomic propositions $AP$ and a partition of $AP$ in inputs $I$ and outputs $O$, the LTL synthesis problem is to determine whether a circuit over inputs $I$ and outputs $O$ exists such that the circuit satisfies the specification. If no such circuit exists, we call the specification to be unrealizable. Typically, an LTL specification is decomposed into assumptions, posed on the inputs from the environment, and guarantees, that determine how to react to the inputs. For training Transformers, we represent assumptions and guarantees as sequences with a tree positional encoding [47]. The basic idea is, to encode the path through the syntax tree for each character. Since LTL has only unary and binary operations, this is encoded by appending either $1, 0$, representing the left child or $0, 1$, representing the right child, in front of the encoding. Figure 2 shows an example tree positional encoding for the response pattern $\square(r \rightarrow \diamondsuit g)$.

The AIGER format became an established format for benchmarks, competitions, and tool implementations in both computer-aided verification and reactive synthesis. The AIGER format represents sequential circuits as and-inverter graphs in both ASCII and binary format. In this work, we refer to the original version 20071012 in ASCII format [4]. The first line in an AIGER file in ASCII format contains the header that is the format identifier string "aag" followed by 5 non-negative integers indicating the maximum variable index $M$, the number of inputs $I$, the number latches $L$, the number of outputs $O$, and the number of AND gates $A$. The header is followed by $I$ lines defining the inputs, $L$ lines defining the latches, $O$ lines defining outputs, and $A$ lines defining the AND gates. An optional symbol table to name inputs, outputs, and latches and a comment section may follow after the definitions. Inputs, latches, outputs, and AND gates are defined using variables and literals represented as non-negative integers. The relationship between literals and variables is that we divide the literal by 2 to obtain the variable and if the literal modulo 2 equals 1 it corresponds to the negated variable and if the literal modulo 2 equals 0 it corresponds to the unnegated variable. Further literal 0 represents the Boolean constant $\bot$ and literal 1 represents the Boolean constant $\top$. Inputs are defined as unnegated literals. Latches are defined as two literals separated by a space. The first literal provides the current state of the latch and the second literal the next state of the latch. Outputs are defined as arbitrary literals. AND gates are defined as three literals separated by a space. The first literal is the output of the AND gate and the second and third literals are the inputs of the AND gate.

For our small arbiter example above, we show below a circuit (left) and its AIGER representation (right), which is actually a prediction of a hierarchical Transformer.

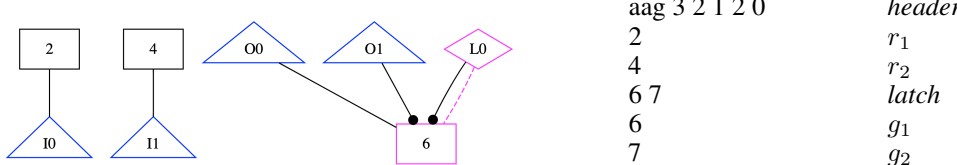

| | | |
|---|---|---|
| aag 3 2 1 2 0 | | *header* |
| 2 | | $r_1$ |
| 4 | | $r_2$ |
| 6 7 | | *latch* |
| 6 | | $g_1$ |
| 7 | | $g_2$ |

The triangles represent inputs and outputs, the rectangles represent variables, the diamond-shaped variables represent latches and the black dots represent inverter (NOT gates). The circuit implementation ignores the inputs $I0$ and $I1$, which represent both requests $r_1$ and $r_2$ (except for unnecessarily assigning them to variables 2 and 4). The circuit implementation satisfies the specification by alternating indefinitely between both outputs $O0$ and $O1$, which represent both grants $g_1$ and $g_2$, independently of the given inputs. This is, in fact, the smallest solution satisfying the simple arbiter specification above. The hierarchical Transformer also predicts correct circuit implementations for more involved specifications where the circuit has to react to inputs (see, for example, Section 5 for an arbiter that prioritizes a certain request).

### 3.2 Data Generation

From the LTL track of SYNTCOMP 2020 [26] we collected 346 benchmarks in *Temporal Logic Synthesis Format* (TLSF) [27]. Using SyFCo [27] we translated the TLSF specifications to the BoSy input format [14]. The BoSy input format is a JSON-based format representing specifications as a list of assumptions and a list of guarantees where assumptions and guarantees can be arbitrary LTL formulas. The LTL specification results from the implication of the conjunction of assumptions to the conjunction of guarantees. An example of the format for a prioritized arbiter specification is shown in Listing 1 in the appendix. From the 346 benchmarks in BoSy input format we collected assumptions and guarantees and filtered LTL formulas with more than five inputs and more than five outputs, which is a restriction that is comparable with most SYNTCOMP specs: 242 of 346 have $\leq 5$ inputs and 274 have $\leq 5$ outputs. Further, we filtered out specifications with an abstract syntax tree of size greater than 25 resulting in 157 instantiated assumption patterns and 1942 instantiated guarantee patterns. In a final step, we renamed inputs and outputs with a uniform random choice from input atomic propositions $i_0, i_1, i_2, i_3, i_4$ and a uniform random choice from output atomic propositions $o_0, o_1, o_2, o_3, o_4$, respectively. The table below, shows three random examples of assumption patterns and three random examples of guarantee patterns.

| assumption patterns | guarantee patterns |
|---|---|
| $\square(i_0 \wedge \bigcirc(\neg o_0 \wedge \neg o_1) \rightarrow \bigcirc i_0)$ | $(o_2 \, \mathcal{U} \, i_3) \vee \square o_2$ |
| $\square \lozenge i_0$ | $\square(i_0 \rightarrow \bigcirc(o_3 \vee i_3 \vee \bigcirc(o_3 \vee i_3 \vee \bigcirc(o_3 \vee i_3))))$ |
| $\square(\neg i_0 \vee o_3 \vee o_2 \vee o_1 \vee o_0 \vee \bigcirc i_0)$ | $\square(\neg o_2 \vee \neg o_4)$ |

Given the set of specification patterns, we generate a dataset for supervised learning, i.e., pairs of specifications and systems, by combining randomly instantiated specification patterns. Specifically, we alternate between sampling guarantees until the specification becomes unrealizable and sampling assumptions until the specification becomes realizable where the number of trials to find a suitable assumption is limited to 5. Further, we implemented stopping criteria that limit the maximal number of guarantees to 10, the maximal number of assumptions to 3, and the runtime for the synthesis tool to 120 seconds. If the resulting specification is unrealizable we also consider its realizable predecessor for our dataset. Apart from that intermediate specifications are discarded. To synthesize specifications, we use the LTL synthesis tool Strix [35]. Systems are represented in the AIGER format. For unrealizable specifications we provide an AIGER circuit representing the winning strategy for the environment, i.e., a counter strategy showing that the specification is unrealizable. When synthesizing specifications, we provide all five inputs $i_0, i_1, i_2, i_3, i_4$ and all five outputs $o_0, o_1, o_2, o_3, o_4$ to the tool such that all AIGER circuits in our datasets have the same five inputs and the same five outputs. Based on the AIGER format, we apply two additional filters when generating data: 1) we filter AIGER circuits exceeding a maximum variable index of 50, 2) we filter AIGER circuits with $k$ AND gates if the number of circuits in the dataset with $k$ AND gates exceeds 20% of the dataset size. This filtering especially reduces the number of circuits that have a low amount of AND gates.

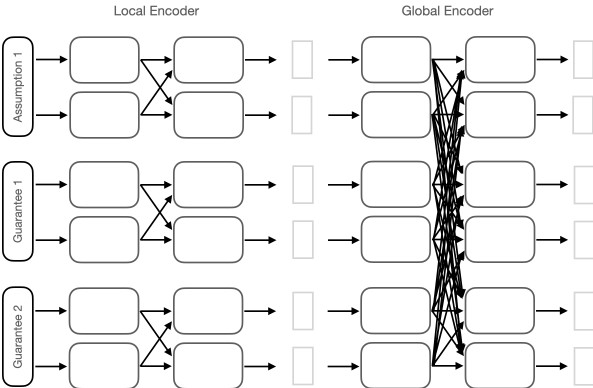

Figure 3: A hierarchical Transformer [33] first encodes assumption and guarantee patterns in isolation, before encoding them globally.

The data generation method allows to generate a large number of specifications from a comparatively small set of specification patterns; especially the generation of specifications that include meaningful assumptions and are realized by complex implementations. Following the described method, we constructed a dataset containing $250\,000$ unique samples split into $200\,000$ training samples, $25\,000$ validation samples, and $25\,000$ test samples. We included the unrealizable specifications met through the first stopping criteria such that half of the dataset consists of unrealizable specifications. Figure 1 shows an example of a realizable specification in the test data.

## 4  Experimental Setup

Based on the code base of DeepLTL [21] (MIT license), we implemented a hierarchical Transformer [33] and augment it with a tree-positional encoding [47].[2] In contrast to a baseline Transformer, the encoder has two types of layers, local and global layers.

The local layers encode individual assumptions and guarantees, and only the global layers can combine the representations of tokens across all assumptions and all guarantees. With this hierarchical encoding, we gain approximately $10\%$ of accuracy across all models compared to using a standard Transformer (see Figure 6). Figure 3 sketches the use of local and global layers in the encoder for our setting.

We trained hierarchical Transformers with model dimension 256. The dimension of the feed-forward networks was set to 1024. The encoder employs 4 local layers followed by 4 global layers, and the decoder employs 8 (unmodified) layers. All our attention layers use 4 attention heads. We trained with a batch size of 256 for $30\,000$ steps and saved the model with the best accuracy per sequence on the validation data. We trained on an NVIDIA DGX A100 system for around 10 hours.

### 4.1  Training Details

The Transformer architecture is a sequence-to-sequence model trained to predict a sequence of output tokens provided a sequence of input tokens. Similarly, we provide multiple sequences of input tokens to an hierarchical Transformer. Assumptions and guarantees are LTL formulas and can thus be directly represented as sequences of tokens with each atomic proposition, Boolean operator, temporal operator, and Boolean constant being a separate token. We omit

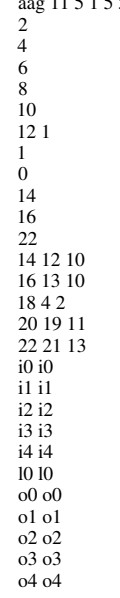

```
aag 11 5 1 5 5
2
4
6
8
10
12 1
1
0
14
16
22
14 12 10
16 13 10
18 4 2
20 19 11
22 21 13
i0 i0
i1 i1
i2 i2
i3 i3
i4 i4
l0 l0
o0 o0
o1 o1
o2 o2
o3 o3
o4 o4
```

Figure 4: AIGER representation of the circuit in Figure 1.

[2]The code, our data sets, and data generators are part of the Python library ML2 (https://github.com/reactive-systems/ml2).

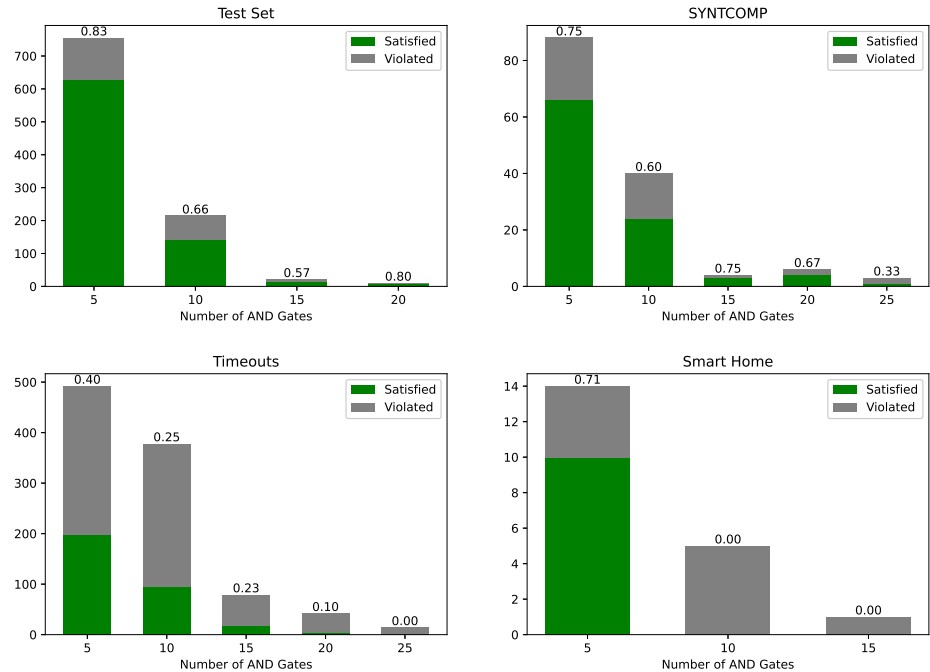

Figure 5: Accuracy with respect to the size of the synthesized circuits measured by the number of AND gates for test set (top, left), SYNTCOMP (top, right), timeouts (bottom, left), and smart home benchmarks (bottom, right). Number of AND gates are binned into intervals of size 5.

parentheses because we add a tree-positional encoding [47] that identifies each token with its position in the abstract syntax tree of the LTL formula (see Figure 2). To distinguish assumptions from guarantees in the global step we prepend assumptions with a special assumption token. Circuits are in AIGER format that we represent as a sequence of tokens by representing each integer with a corresponding token and replacing each newline character with a special new line token. Since all circuits in our dataset have the same inputs and outputs we can omit the header and the symbol table when tokenizing an AIGER circuit. Additionally, we include a special realizability token at the beginning of the sequence indicating whether a specification is realizable.

We trained all models using the Adam optimizer [29] with $\beta_1 = 0.9$, $\beta_2 = 0.98$ and $\epsilon = 10^{-9}$. The optimizer was used with a learning rate schedule proposed by Vaswani et al. [51] that increases the learning rate linear for a given number of *warmup steps* followed by a decreasing learning rate proportionally to the inverse square root of the step number. In our experiments, we used 4000 warmup steps as proposed by Vaswani et al. [51].

## 4.2 Performance Measures

There are infinitely many circuits satisfying a realizable LTL specification. To evaluate the performance of the trained models we thus distinguish between the *syntactic accuracy* and the *semantic accuracy*: For a dataset of specifications and systems, the syntactic accuracy measures the percentage of the Transformer's predictions that match the circuit in the dataset. Potentially, a prediction that does not match the system still satisfies the specification. We thus also measure the semantic accuracy, i.e., the percentage of the Transformer's predictions that satisfy the specification. Note that, when using a beam search algorithm only one of the predictions needs to match the system in the dataset or satisfy the specification, respectively. To model check predictions we use the nuXmv model checker [10]. When training Transformers on the (easier) LTL trace generation problem [22], a significant difference between syntactic and semantic accuracy was observed. It appears that the Transformers rather generalize to the semantics of the logic than the particularities of the data generator. As we will see in the next section, our results are consistent with this observation even for the "harder" problem of predicting circuits.

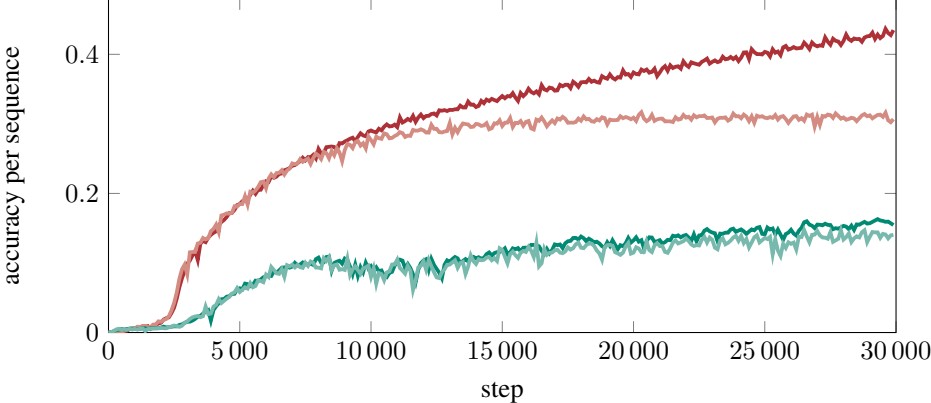

Figure 6: Accuracy per sequence over the training course shown for the training split (red) and validation split (light red) when training the hierarchical Transformer and for the training split (green) and the validation split (light green) when training the standard Transformer.

## 5 Experiments

In this section, we report on a variety of experiments that analyze the performance of hierarchical Transformers on the circuit synthesis task and their generalization behavior. In the following, we will first analyze the overall performance of the models and see that they often construct different solutions, yet correct ones, than the classical tool we generated the training data with. For this, we consider four different test sets and group results on the size of the predicted circuits. Secondly, we compare the training with our data mining method against the ground truth, i.e., against a training of a hierarchical Transformer on the raw SYNTCOMP benchmarks. Thirdly, we compare the models performance on realizable and unrealizable specifications, where for the latter the model is supposed to construct a circuit representing a counter strategy. Lastly, we will take a deeper look into one of the specifications, which, compared to the example in Section 3 is an arbiter that prioritizes a certain request.

**Overall results.** We tested our models on four different datasets. A `Testset` consisting of held-out instances generated by our data mining method, the `SYNTCOMP` set, consisting of the synthesis competition benchmarks, a set `Timeouts` that consists of generated specifications on which Strix, the classical synthesis tool that we used for generating the circuits, timed out ($< 120s$), and an out-of-distribution (OOD) benchmark set `Smart Home` consisting of specifications for smart homes. We consistently observed in all experiments that the beam search significantly increases the accuracy. When analyzing the results we found that the beam search often yields several correct circuits. For a beam size of $16$ and the `Testset`, on average $4.6$ of the $16$ AIGER circuits satisfy the specification.

In our `Testset` (see Table 1), we observe in many cases that the circuit prediction of our model is different from the circuit the tool would synthesize. Since it has already shown that this gap between syntactic and semantic accuracy exists for such tasks [22], we concentrate on the semantic accuracy, i.e., the total accuracy. When analyzing the size of those circuits, we found both smaller and larger

| Dataset | Beam Size 1 | Beam Size 4 | Beam Size 8 | Beam Size 16 |
|---------|-------------|-------------|-------------|--------------|
| Testset | $53.6(31.1) \pm 2.4$ | $70.4(39.0) \pm 2.3$ | $75.8(41.9) \pm 2.1$ | $79.9(44.5) \pm 2.0$ |
| SYNTCOMP | $51.9 \pm 2.2$ | $60.0 \pm 1.5$ | $63.6 \pm 1.9$ | $66.8 \pm 1.2$ |
| Timeouts | $11.7 \pm 1.1$ | $21.1 \pm 0.9$ | $25.9 \pm 1.0$ | $30.1 \pm 1.2$ |
| Smart Home | $22.9 \pm 3.6$ | $31.4 \pm 7.1$ | $44.8 \pm 6.5$ | $40.0 \pm 6.5$ |

Table 1: Accuracy reported on test data, SYNTCOMP benchmarks, timeouts, and smart home benchmarks for different beam sizes averaged over 5 trainings including standard deviation. For the test data we show the syntactic accuracy in parenthesis.

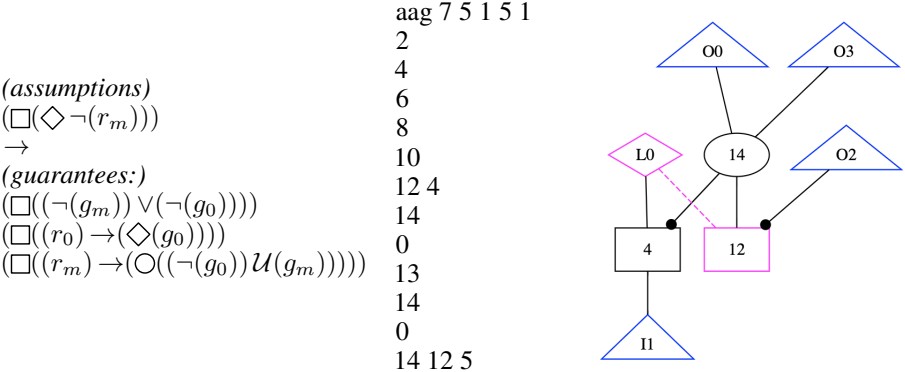

*(assumptions)*
$(\Box(\Diamond\neg(r_m)))$
$\rightarrow$
*(guarantees:)*
$(\Box((\neg(g_m))\vee(\neg(g_0))))$
$(\Box((r_0)\rightarrow(\Diamond(g_0))))$
$(\Box((r_m)\rightarrow(\bigcirc((\neg(g_0))\,\mathcal{U}(g_m)))))$

```
aag 7 5 1 5 1
2
4
6
8
10
12 4
14
0
13
14
0
14 12 5
```

Figure 7: The specification (left), the predicted AIGER circuit (middle) and the visualization of the circuit (right) for a prioritizing arbiter.

circuits, with no significant decrease or increase in average circuit size. In total, the model was able to solve $79.9\%$ of the held-out generated test instances with a beam size of 16.

While the training data is based on specification patterns extracted from SYNTCOMP benchmarks it is unlikely that our data generation process reassembles SYNTCOMP benchmarks. This allows to evaluate the model on them. After filtering out benchmarks with more than 5 inputs/outputs, more than 12 properties, and properties of size greater than 25, the model achieved an accuracy of $66.8\%$ for the resulting 145 benchmarks using a beam size of 16.

For a timed out specification it is not known whether it is realizable or unrealizable. The model achieves an accuracy of $30.1\%$ for beam size 16 demonstrates that our approach can yield performance gains in practice. To highlight the capabilities of our model we display in Figure 9 in the appendix the largest circuit that is predicted for a timed out specification and satisfies the specification.

We constructed the `Smart Home` set, with the same restriction as for SYNTCOMP, from a recently published benchmark set for synthesizing smart home applications [1]. The hierarchical Transformer is able to solve $44.8\%$ of the provided instances. When compared to the full benchmark (i.e., without the size restrictions), the model solved $11.1\%$ of the formulas. Note that this benchmark set was not used to mine specifications from and the benchmarks include instances with larger assumptions and guarantees than seen during training.

We also analyzed the performance of the model depending on the size of the predicted circuit. Results are shown in Figure 5. As expected, for larger circuit implementations, the model accuracy drops. The size distribution of the training data resembles the size distribution of the test set (top left in Figure 5 and Figure 8). Meaning that the model has seen a significantly lower percentage of large circuits during training. Future experiments have to determine how large the training data and the hierarchical Transformers could be scaled, before the training process breaks down.

**Training on raw SYNTCOMP benchmarks.**  We did a baseline experiment by training with various batch sizes on the raw SYNTCOMP benchmark. This training (not surprisingly) fails whereas our data generation method enables a stable training.

**Unrealizabile Specifications.**  The training data contains both realizable and unrealizable specifications. In Table 3 we analyze the accuracy for realizable and unrealizable specifications separately on our test data. While the syntactic accuracy is higher for realizable specifications, in terms of the

|  | Beam Size 1 | Beam Size 4 | Beam Size 8 | Beam Size 16 |
|---|---|---|---|---|
| Realizable | 50.8 (39.0) | 64.3 (48.0) | 67.5 (50.0) | 70.7 (52.6) |
| Unrealizable | 55.4 (23.0) | 74.6 (31.9) | 81.0 (35.2) | 86.7 (39.0) |

Table 2: Accuracy on `Testset` reported separately for realizable and unrealizable specifications. For different beam sizes we report the semantic accuracy and the syntactic accuracy in parenthesis.

semantic accuracy the model solves unrealizable specifications more accurately. Further, we found for a beam size of $1$ that the Transformer predicts the correct realizability token for $91.4\%$ of the specifications from the test data.

**Prioritizing arbiter.** Building on the example of Section 3, we show that the model can handle more interesting, real-world specifications. Figure 7 shows the specification, AIGER file and the circuit visualization of an arbiter that prioritizes one of the requests whenever access is requested by both processes at the same time; meaning that the implementation can no longer ignore the input as for the example in Section 3.

## 6 Conclusion

We proposed a method to address the lack of data for training a neural network on the task of synthesizing circuits out of LTL specifications. We mine specification patterns from the annual reactive synthesis competition (SYNTCOMP) and generate new formulas by combining multiple specification patterns. We showed that this dataset can be used to successfully train hierarchical Transformers on the LTL synthesis problem for specifications composed of specification patterns. We also showed that the models generalize to unseen specifications, including specifications that are both realizable and unrealizable and specifications that cannot be solved by a classical synthesis tool within a time limit of $120$ seconds. Furthermore, we performed an out-of-distribution test on a recently added benchmark set on synthesis problems for smart homes. Experimental results suggest that the Transformer can be especially useful for predicting unrealizability.

## Acknowledgments and Disclosure of Funding

This work was partially supported by the European Research Council (ERC) Grant OSARES (No. 683300) and the Collaborative Research Center "Foundations of Perspicuous Software Systems" (TRR 248, 389792660).

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
