## Appendix

For a given set of atomic propositions $AP$, the syntax of LTL formulas over $AP$ is defined as:

$$\varphi, \psi ::= \top \mid a \mid \neg\varphi \mid \varphi \wedge \psi \mid \bigcirc\varphi \mid \varphi\,\mathcal{U}\,\psi \ ,$$

where $\top$ is the Boolean constant, $a \in AP$, $\neg$ and $\wedge$ are the Boolean connectives and $\bigcirc$ and $\mathcal{U}$ are temporal operators. We refer to $\bigcirc$ as the *next* operator and to $\mathcal{U}$ as the *until* operator. Other Boolean connectives can be derived. Further, we can derive temporal modalities such as *eventually* $\Diamond\varphi := \top\,\mathcal{U}\,\varphi$ and *globally* $\Box\varphi := \neg\Diamond\neg\varphi$. For a given set of atomic propositions $AP$, the semantics of an LTL formula over $AP$ is defined with respect to the set of infinity words over the alphabet $2^{AP}$ denoted by $\left(2^{AP}\right)^\omega$. The semantics of an LTL formula $\varphi$ is defined as the language $Words(\varphi) = \{\sigma \in \left(2^{AP}\right)^\omega \mid \sigma \models \varphi\}$ where $\models$ is the smallest relation satisfying the following properties:

$$
\begin{aligned}
&\sigma \models \top \\
&\sigma \models a && \text{iff } a \in A_0 \\
&\sigma \models \neg\varphi && \text{iff } \sigma \not\models \varphi \\
&\sigma \models \varphi \wedge \psi && \text{iff } \sigma \models \varphi \text{ and } \sigma \models \psi \\
&\sigma \models \bigcirc\varphi && \text{iff } \sigma[1\ldots] \models \varphi \\
&\sigma \models \varphi\,\mathcal{U}\,\psi && \text{iff } \exists j \geq 0.\ \sigma[j\ldots] \models \psi \text{ and } \forall 0 \leq i < j.\ \sigma[i\ldots] \models \varphi
\end{aligned}
$$

where $\sigma = A_0 A_1 \ldots \in (2^{AP})^\omega$ and $\sigma[i\ldots] = A_i A_{i+1} \ldots$ denotes the suffix of $\sigma$ starting at $i$.

Listing 1: Specification of a prioritized arbiter in BoSy input format that is part of the 2020 SYNT-COMP benchmarks [26].

```
{
  "semantics": "mealy",
  "inputs": [
    "r_m",
    "r_0"
  ],
  "outputs": [
    "g_m",
    "g_0"
  ],
  "assumptions": [
    "(G (F (! (r_m))))"
  ],
  "guarantees": [
    "(true)",
    "(G ((! (g_m)) || (! (g_0))))",
    "(G ((r_0) -> (F (g_0))))",
    "(G ((r_m) -> (X ((! (g_0)) U (g_m)))))"
  ]
}
```

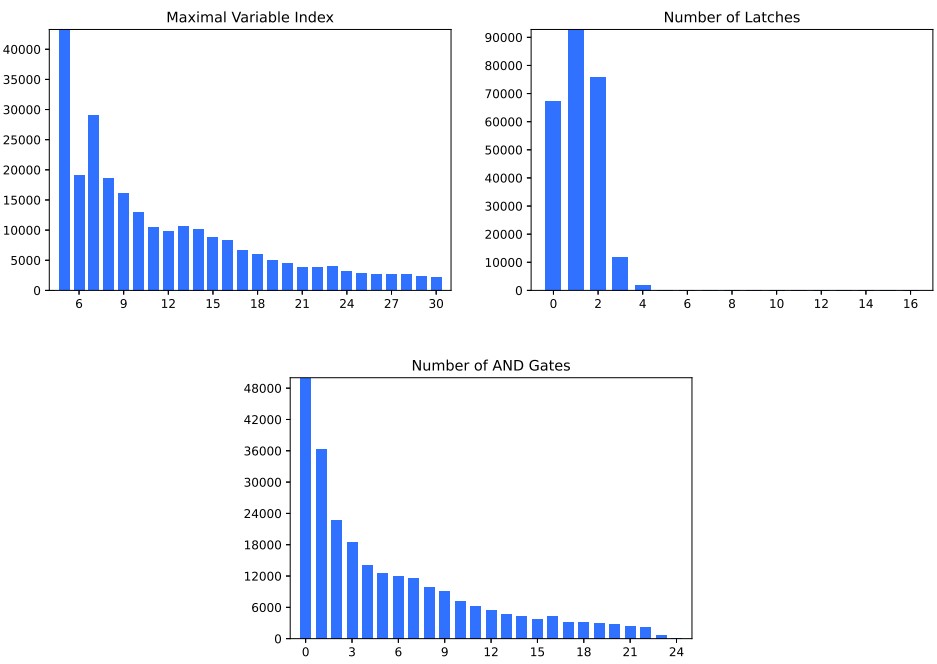

Figure 8: Distribution of maximal variable index, number of latches, and number of AND gates in the dataset.

| $d_m$ | $d_{ff}$ | $n_{loc}$ | $n_{glob}$ | $n_{dec}$ | $n_{heads}$ | Beam Size 1 | Beam Size 16 |
|------|------|------|------|------|------|------|------|
| 256 | 1024 | 4 | 4 | 8 | 4 | 51.6 (28.9) | 81.3 (39.8) |
| 128 | 512 | | | | | 50.7 (28.4) | 76.6 (40.6) |
| 128 | 512 | 2 | 2 | 4 | | 50.3 (28.0) | 76.6 (42.7) |
| 256 | 256 | | | | | 54.5 (30.6) | 81.5 (43.8) |
| 256 | 512 | | | | | 53.4 (30.9) | 78.6 (44.5) |
| 512 | 512 | | | | | 23.3 (4.9) | 57.4 (26.5) |
| | | 2 | 2 | 4 | | 52.8 (30.9) | 79.0 (43.1) |
| | | 2 | 2 | | | 50.6 (27.9) | 77.1 (40.5) |
| | | 2 | 6 | | | 49.9 (25.4) | 79.1 (41.2) |
| | | 3 | 3 | 6 | | 50.5 (28.9) | 76.8 (40.0) |
| | | | | 4 | | 53.8 (30.4) | 78.0 (42.0) |
| | | 5 | 5 | 10 | | 15.8 (4.6) | 45.9 (18.4) |
| | | 6 | 2 | | | 46.2 (27.3) | 74.1 (41.0) |
| | | | | | 8 | 55.3 (31.5) | 78.9 (45.0) |
| | | | | | 16 | 53.6 (30.3) | 78.0 (44.5) |

Table 3: Hyper-parameter search for parameters embedding dimension $d_m$, feed-forward network dimension $d_{ff}$, number of local encoder layers $n_{loc}$, number of global encoder layers $n_{glob}$, number of decoder layers $n_{dec}$, and number of attention heads $n_{heads}$. Empty cells have the same value as the base model (first row). For each choice we report the accuracy on `Testset` for beam size 1 and beam size 16 with syntactic accuracy in parenthesis.

```
aag 21 5 2 5 14
2
4
6
8
10
12 17
14 43
17
0
0
19
0
16 15 12
18 15 13
20 13 8
22 15 9
24 23 21
26 25 7
28 6 3
30 28 20
32 31 27
34 33 5
36 4 3
38 36 7
40 38 20
42 41 35
```

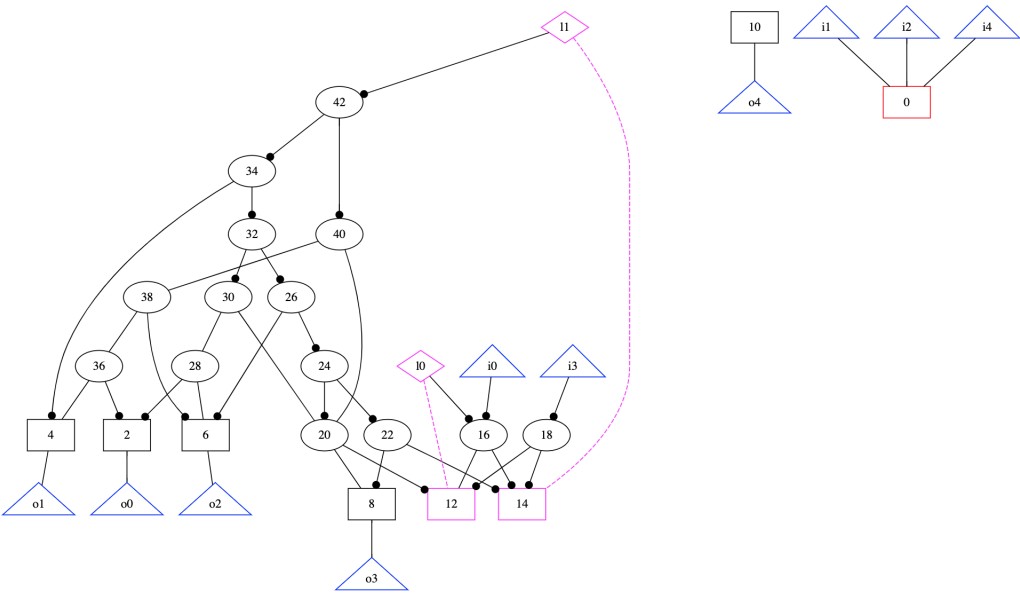

Figure 9: The largest circuit that satisfies a specification on which the classical tool times out.