# OpenReview forum: "Neural Circuit Synthesis from Specification Patterns"
_NeurIPS.cc/2021/Conference — NeurIPS 2021 Poster_

### Official Review · Reviewer_fNW6 · 2021-07-07

**Rating:** 6
**Confidence:** 4

**Summary:**

The paper introduces a deep learning approach for converting LTL specifications to circuits, effectively solving the classic LTL synthesis problem. The architecture is based on a pre-existing transformer setup that targets a representation of the circuit that allows for sequential generation of the graphical representation (AIGER).

A key contribution of the work is the dataset generated by the authors for the task of synthesis. While there is a regularly run contest, the benchmarks are limited in size compared to what deep learning approaches would require. The data stems from random recombinations of fragments found in the existing benchmark sets.

**Ethical Concerns:**

I see no ethical concerns with the work presented.

**Limitations And Societal Impact:**

I see no negative societal impact of the work, and the limitations that stem from the paper are discussed above.

**Main Review:**

## Originality
The work follows a similar architecture to previous work dealing with similar data. However, this seems to be the first approach using those methods to solve the problem of synthesis.

## Quality
The presented work is technically sound, however, there is limited attention given to some of the core limitations:

- The augmentation used for generating the data might make the new LTL fairly meaningless. Without a direct correspondence to LTL that is meaningful (i.e., of the form we would like to synthesize in practice), then the empirical results are less relevant.

- Related, there seems to be quite a bit of overlap between the train/test data sets for much of the results. Even if the precise combination of LTL fragments are not shared, the fragments themselves almost certainly are.

- Much of the analysis hinges on state-of-the-art solvers timing out. At 10 seconds, however, this seems very limited. The low amount should be justified, and the time required for the presented work (i.e., using the hierarchical transformers) should be listed as well.

- The most relevant comparison listed is with the smart home benchmark -- other ones have too much of an overlap between the training and testing sets (see comment above on the fragments). A better analysis might be to train on some SYNTCOMP domains and test on the remainder.

- With only 14% of the smart home solved, compared to 42.9% given the restrictions, demonstrates that the generality of this approach may be a serious limitation.


## Clarity
The paper is written very well, and the only suggestions I have are to address key questions that came to mind while reading:

- What is the justification for the existing filters on the problems? I.e., circuits > 50vars and the computation on circuit AND gates exceeding 20% of the dataset size.

- What is the time spent on each of the beam sizes in computing a solution?

- Judging by the table, the smart home dataset continually improves. Does this trend continue for beam size = 32 or higher?

- Why do unrealizable benchmarks become easier with larger beam sizes?

A final comment is that it may be worth relating the data processing you do to the traditional data augmentation technique in the ML field (e.g., cropping for image-based data). You have, in essence, introduce a domain-specific form of data augmentation to amplify the amount of available data for the task of LTL synthesis.


## Significance
The significance of the results is somewhat difficult to ascertain. On the one hand, there seems to be reasonable coverage for both realizable and unrealizable inputs. But, crucially, how does the proposed approach compare with state-of-the-art techniques for LTL synthesis? We have a sense of how long Strix requires, but not for the proposed approach at inference time. Is there some way to use the invalid circuits the proposed approach produces as a seed to a complete algorithm, and would this be competitive?

I understand that the authors point out in the checklist and conclusion that the aim was _not_ to compete with classical tools, but rather just to address the lack of data for ML methods in the space. However, much of the paper is dedicated to the _use_ of the data, rather than the data itself, and if the performance is really poor then the significance is reduced.

The 88.3% accuracy on realizability classification seems like a really nice result. I would be curious to know if this improves with greater beam sizes (by taking a majority vote, or similar, among those in the beam).

Finally, a major limitation holding back the significance of what's proposed is the size limits placed on things. How might the approach be generalized to arbitrary sizes?

**Time Spent Reviewing:**

6

---

> ### Author Response · Authors · 2021-08-09
> **Reply to Reviewer fNW6**
>
> We thank the reviewer for their review and helpful questions and suggestions to improve this submission!
>
>
>
> - “The augmentation used for generating the data might make the new LTL fairly meaningless. Without a direct correspondence to LTL that is meaningful (i.e., of the form we would like to synthesize in practice), then the empirical results are less relevant.”
>
> The data generation method follows the structure of typical LTL synthesis specs. They consist of a list of assumptions and a list of guarantees (see [26]) of repetitive patterns (see [12] for example). The idea of using patterns and combining them is not new ([12,13,23,38] and around 2000 patterns are enough to specify almost all SYNTCOMP benchmarks). An indicator that the circuits resulting from this data generation are meaningful is that they have a non-trivial amount of and-gates.
>
> - “Related, there seems to be quite a bit of overlap between the train/test data sets for much of the results. Even if the precise combination of LTL fragments are not shared, the fragments themselves almost certainly are.”
>
> The difficulty of the problem lies not in the individual LTL fragments/patterns, but in the interaction between multiple patterns. With about 2K patterns mined from SYNTCOMP, the number of combinations is large enough to define a reasonable train/test split. To validate that the model does not merely overfit, we also tested on the smart home formulas (which are quite different from the other SYNTCOMP formulas).
>
> - "Much of the analysis hinges on state-of-the-art solvers timing out. At 10 seconds, however, this seems very limited. The low amount should be justified, and the time required for the presented work (i.e., using the hierarchical transformers) should be listed as well."
>
> The timeout was chosen to be higher than the model's prediction time - so that for those formulas, the Transformer would indeed improve over Strix. Meanwhile we repeated most of our experiments also with a 120 second timeout for Strix (both for generating training data and test data). The results remain very similar and we will list them in the appendix.
>
> - “What is the justification for the existing filters on the problems? I.e., circuits > 50vars and the computation on circuit AND gates exceeding 20% of the dataset size.”
>
> Increasing the number of inputs and outputs has a non-obvious effect on the training data: The likelihood that two specification patterns share inputs and outputs decreases, making the resulting problems simpler on average. So we followed [22] and (arbitrarily) set the bound to 5. The restriction is comparable with most SYNTCOMP specs: 242 of 346 have <= 5 inputs and 274 have <= 5 outputs, but this is - obviously - a restriction that we want to lift in the future. Additionally, filtering out circuits when already 20% of the circuits have k AND-gates also makes the dataset harder. Without filtering the dataset would contain more circuits with 0 AND-gates. We will clarify this in the paper.
>
> - What is the time spent on each of the beam sizes in computing a solution?
> - Judging by the table, the smart home dataset continually improves. Does this trend continue for beam size = 32 or higher?
>
> We have not tested the performance in these cases, but will add those numbers to the paper.
>
> - Why do unrealizable benchmarks become easier with larger beam sizes?
>
> We are not sure why the unrealizable benchmarks profit slightly more from larger beam sizes. We suspect that the asymmetric nature of the data generation process makes the unrealizable specifications a little easier than the realizable specifications.
>
> - “A final comment is that it may be worth relating the data processing you do to the traditional data augmentation technique in the ML field (e.g., cropping for image-based data). You have, in essence, introduced a domain-specific form of data augmentation to amplify the amount of available data for the task of LTL synthesis.”
>
> We will point this out in related work.
>
> - "But, crucially, how does the proposed approach compare with state-of-the-art techniques for LTL synthesis? We have a sense of how long Strix requires, but not for the proposed approach at inference time. Is there some way to use the invalid circuits the proposed approach produces as a seed to a complete algorithm, and would this be competitive?"
>
> Strix solves most of the problems in the Syntcomp benchmark and the goal of this paper was not to compete with Strix (we will make this clearer in the paper). Your suggestion is leading directly to our plans for future work. One of many possibilities is to utilize so-called repair algorithms. We envision, with this line of work, to become competitive or at least augment the formal methods tools with neural heuristics. For example, although Neurosat [46] was not competitive, it was also successfully used for unsat core predictions [45].
>
>
> - "The 88.3% accuracy on realizability classification seems like a really nice result. I would be curious to know if this improves with greater beam sizes (by taking a majority vote, or similar, among those in the beam)."
>
> Thanks for the suggestion! We are running experiments and will include it in the paper.
>
>
> - "Finally, a major limitation holding back the significance of what's proposed is the size limits placed on things. How might the approach be generalized to arbitrary sizes?"
>
> As detailed above, the size restrictions have nontrivial interactions with the hardness of the datasets. To encourage generalization to larger formulas (e.g. with more input/output variables) we would have to carefully control the amount of interactions between the different patterns in each formula. We believe that this process should be driven by applications.

---

> > ### Comment · Reviewer_fNW6 · 2021-08-12
> > **Rebuttal Response**
> >
> > Thank you for the response. I think the added evaluations would make for a stronger paper -- if you have any preliminary results to share here, then please do.
> >
> > On the topic of random LTL creation, I don't feel that it's enough to just have the same characteristics/statistics as existing formulae. It feels akin to random SAT instances that may follow the same distribution of clause-variable ratios, but lack the true structure of industrial instances. I think the evaluation with the smart home instances is the most demonstrative as a consequence of this.

---

> > > ### Author Response · Authors · 2021-08-17
> > > **Reply**
> > >
> > > Some experiments have finished already. We will provide results to other experiments as soon as possible.
> > >
> > > - Here are results for the case where we increase the timeout of the synthesizer to 120s (instead of 10s) averaged over 5 runs for beam sizes 1, 4, 8, and 16 with standard deviation in parentheses:
> > >
> > > Dataset | Beam Size 1 | Beam Size 4 | Beam Size 8 | Beam Size 16
> > > --- | --- | --- | --- | ---
> > > Testset | 53.64 (2.35) | 70.44 (2.25) | 75.75 (2.13) | 79.91 (2.04)
> > > Syntcomp | 51.86 (2.20) | 60.0 (1.45) | 63.59 (1.92) | 66.76 (1.19)
> > > Timeouts | 11.70 (1.07) | 21.05 (0.90) | 25.88 (0.96) | 30.12 (1.16)
> > > Smart Home | 22.86 (3.56) | 31.43 (7.13) | 44.76 (6.46) | 40.0 (6.46)
> > >
> > > - The performance increases for the smart home benchmarks with even larger beam sizes. Averaged over 5 runs for beam sizes 32, 64, and 128 with standard deviation in parentheses the numbers are: 47.62 (6.73), 59.05 (4.86), and 57.14 (3.01).
> > >
> > > Our approach is to generate training data from mined patterns to avoid this problem with random formulas. The mined patterns somewhat represent the "structure" of the formulas, which is supported by the good performance on SYNTCOMP and the smart home benchmarks.

---

### Official Review · Reviewer_Z9R3 · 2021-07-14

**Rating:** 5
**Confidence:** 4

**Summary:**

This paper applies hierarchical Transformers to the linear-time logic (LTL) synthesis problem. The contributions are two-fold: 1) data (or LTL specification) generation, which is achieved by conjoining specification patterns from existing SYNTCOMP benchmarks; 2) empirical evaluations of hierarchical Transformer on synthetic specifications as well as the original SYNTCOMP benchmarks.

**Ethical Concerns:**

There are no ethical issues since this paper concerns a technical circuit synthesis problem.

**Limitations And Societal Impact:**

The authors do not explicitly discuss their limitations or potential negative societal impact. There is no concern about the latter.

Dataset could be more challenging without artificial filters, which would thus be more useful to inspire future ML-based methods for LTL synthesis. It is also great to see the performance of more one than ML methods on this newly created dataset.

**Main Review:**

### Contribution and Significance
Transformers have been applied for the same LTL synthesis problem recently by Hahn et al. 2021 [22]. And this work applies a hierarchical transformer developed by Li et al. 2021 [33].  So both the problem and used techniques are already known. The main contribution of this work is the development of a new LTL synthesis dataset. Specifically, all synthesized LTL specifications share the same form: $A \rightarrow G$, where $A$ is a set of assumptions and $G$ is a set of guaranttees; $A$ and $G$ are randomly chosen from existing benchmark suite used in the LTL track of SYNTCOMP. This dataset would be useful for evaluating machine learning-based methods for LTL synthesis.

### Clarity and Quality
This paper is well-written. It gives a nice introduction to LTL synthesis problem and systemtically summarizes both recent work on using neural architectures for logical reasoning and many classic approaches. The data generation and data format are presented in a very clear manner.  The experimental evalution is also well-done, assuming the success criteria is showing a particular existing method on the newly generated dataset works well, which is something I hesitate to agree with.

The authors appear to carefully calibrate the data generation process so that hierarchical Transformers perform reasonally well. In terms of dataset generation (the main contribution of this work), I don't see a compelling reason why specifications with more than five inputs or outputs have to be filtered away. Similarly, why circuits with more than 20% k-AND gates should be filtered? Making the dataset easy to solve is not a convincing argument. On the contrary, keeping some challenging instances in a dataset woud make it more valuable.

### Questions
Q1: Normal LTL specifications encode certain safety or liveness property a system should possess. Could (and how does) randomly combining assumptions with guaranttees end up with "meaningful" LTL specifications?

Q2: A LTL specification like $A \rightarrow G$ will be trivially satisfied when its assumptions $A$ are not met. Such kind of solutions may not be very interesting. Essentially, there are two kinds of solution for a specification $A \rightarrow G$: first, a solution to $\neg A$; and second, a solution to $A \wedge G $.   How often does each situation happens when a solution is found by the neural model?

Q3: From the shared GitHub repo, Smart Home seems to be part of the SYNTCOMP benchmarks, instead of independent one. Can you clarify that?

Q4: What is the guarantee on unrealizable specifications? Does the model always produce a counter strategy?

Q5: Counter strategy sounds really interesting, but the paper only mentions it in two sentences with explaining how it is achieved. How is it fundamentally different (e.g. requiring a different method or training process) from a winning strategy ?

**Time Spent Reviewing:**

7.5

---

> ### Author Response · Authors · 2021-08-09
> **Reply to Reviewer Z9R3**
>
> We thank the reviewer for their review and helpful questions!
>
> Unfortunately, we believe there has been a misunderstanding about the problem we consider in our paper:
>
>
> - “Transformers have been applied for the same LTL synthesis problem recently by Hahn et al. 2021 [22]”
>
> Hahn et al. 2021 [22] consider the trace generation problem, while we consider the circuit synthesis problem, which is much harder (2-EXPTIME vs PSPACE). The output of their method is a single trace, while in our work, the predicted output is a circuit computing a correct trace for every possible sequence of inputs.
>
> - “I don't see a compelling reason why specifications with more than five inputs or outputs have to be filtered away. Similarly, why circuits with more than 20% k-AND gates should be filtered? Making the dataset easy to solve is not a convincing argument. On the contrary, keeping some challenging instances in a dataset woud make it more valuable.”:
>
> Increasing the number of inputs and outputs has a non-obvious effect on the training data: The likelihood that two specification patterns share inputs and outputs decreases, making the resulting problems simpler on average. So we followed [22] and (arbitrarily) set the bound to 5. The restriction is comparable with most SYNTCOMP specs: 242 of 346 have <= 5 inputs and 274 have <= 5 outputs, but this is - obviously - a restriction that we want to lift in the future. Additionally, filtering out circuits when already 20% of the circuits have k AND-gates also makes the dataset harder. Without filtering the dataset would contain more circuits with 0 AND-gates. We will clarify this in the paper.
>
> Answers:
>
> - “Normal LTL specifications encode certain safety or liveness property a system should possess. Could (and how does) randomly combining assumptions with guarantees end up with "meaningful" LTL specifications?”
>
> Note that the specification patterns we use to compose specifications encode safety and liveness properties! In fact, they are sufficient to encode the majority of practical specifications (see [12,13,23,38] and around 2000 patterns are enough to specify almost all SYNTCOMP benchmarks).
>
> - “A LTL specification like $A \rightarrow G$ will be trivially satisfied when its assumptions A are not met. Such kind of solutions may not be very interesting. Essentially, there are two kinds of solution for a specification $A \rightarrow G$ first, a solution to not $A$; and second, a solution to $A$ and $G$. How often does each situation happens when a solution is found by the neural model?”
>
> We use assumption patterns sparsely. The data generation process first samples guarantee patterns. Only if the specification becomes unrealizable, assumption patterns are sampled, trying to make it realizable again ($A$ can be empty). This ensures that assumptions are really needed.
>
> - “From the shared GitHub repo, Smart Home seems to be part of the SYNTCOMP benchmarks, instead of an independent one. Can you clarify that?”
>
> The Smart Home benchmarks have only recently been added to the GitHub repository. They were not part of the SYNTCOMP 2020 benchmarks when we started working on this project.
>
>
> - “What is the guarantee on unrealizable specifications? Does the model always produce a counter strategy?”
>
> The model has to produce a correct counter strategy.
>
>
> - “Counter strategy sounds really interesting, but the paper only mentions it in two sentences with explaining how it is achieved. How is it fundamentally different (e.g. requiring a different method or training process) from a winning strategy ?”
>
> The LTL synthesis problem can be thought of as a 2-player game between the environment and the implementation. In a counter strategy, the environment chooses inputs in such a way that the implementation will always lose. For example, for the specification $(G~i) \rightarrow false$, meaning if input $i$ is enabled all the time something bad will happen, the counter strategy is to enable $i$ all the time.

---

> > ### Comment · Reviewer_Z9R3 · 2021-08-20
> > **Thanks for your response**
> >
> > Thank you for clarifying the dataset and some guarantees of the proposed approach. Although the evaluation is still a bit unsatisfactory, I am positive about this work. I hope the authors will incorporate responses here and responses to other reviewers in the next revision.

---

### Official Review · Reviewer_jHAd · 2021-07-17

**Rating:** 8
**Confidence:** 4

**Summary:**

The paper proposes a transformer-based NN methodology for predicting hardware circuits given an LTL specification. This is a well-studied problem in model-checking, so this work represents another attempt of using NNs to solve tasks normally solved using reasoning/symbolic techniques.

The general idea is to cast the problem as a sequence to sequence problem and use a variation of the widely-used transformer architecture. As in many other cases, data-driven approaches for reasoning tasks produce approximations with no guarantees of correctness. That can be useful in some cases but is solving a different problem. In this case, it is possible to verify that the design satisfies the LTL specification. So, the whole approach is incomplete but sound: a better proper than an approximation.

Another risk of this kind of work is a naive use of observational data in combinatorial domains like this one that leads to unsupported claims on generalization. A typical methodological error works as follows. The data is split randomly. If results are good in the test set, then it is claimed that the method is solving the problem. The problem is that conclusion. A test set sampled from the original dataset might have many share many constituents with the train set, so the test set might be similar to the training set. That doesn't mean, of course, that the model is not significant; just that the experiments don't support that.

In this paper, the dataset was build using data from a model-checking competition but combining two parts of the problem (assumptions and guarantees) to get more combinations. That means that almost all assumptions and guarantees will be shared between the training and the test set. However, the evaluation includes a test split from that dataset, as well as three other datasets. These three datasets support the claims of the paper.

There is an additional risk common in this kind of work: not comparing with the state-of-the-art. While this paper does not compare with them directly, the three extra datasets include one where Strix, a model checking tool, timeout after 10 seconds. The 10s seconds is arbitrary, but it shows that some problems are non-trivial. In all fairness, Strix see those instances for the 1st time, so it might deserve more time.

Main question:

- What if Strix is given, let's say, two hours for attempting to solve the example of Fig 9, of the supplementary material?

- A sequence to sequence model does not have, in principle, limitations in the size of the input and output. However, it's entirely possible -even likely– that a model checking tool might be more robust in a subclass of simpler for larger problems. I noticed that table 1 reports "68.3%" accuracy for beam-size 16, SYNTCOMP but *after* "ﬁltering out benchmarks with more than 5 inputs/outputs, more than 12 properties, and properties of size greater than 25".  **Did you test in any instance out of this restriction? For instance, what's the accuracy of the rest of the SYNTCOMP formulas? Please provide separated numbers**. I find the absence of this result a bit surprising. If the results were good, please add them to the paper. Otherwise, add this as a limitation of the method.

- I understand this is not introducing a new architecture but using an existing one. While the code is provided, the description of the parameters is extremely brief. For instance, on page 6, what does "model dimension" mean? Why this architecture? I think these deserve a better explanation. Additional details and discussions can be added in the supplementary material. If you need more space, compact fig 2 and 6.

These are the most important issues. Their answer might affect my recommendation.

For now, I must conclude the contributions of the paper are significant for the task, so I recommend its weak acceptance. If the answers were positive, I'd increase my score.

----

Thank you for your response. I'm more convinced now about the merits of the paper. I'm reviewing my scores accordingly.

There are have been many attempts to solve combinatorial problems using deep learning. In my opinion, some are more famous than they should be like an early one on the TSP. Many of those papers share typical tricks that make the problem easier:

- Change the problem to a softer evaluation. For instance, one thing is getting a tour in a graph, something else getting a good one, it's far harder to get the optimal one.
- Not being able to scale to bigger problems.

This paper has issue (2), but it is transparent about it.

Now, for issue (1), they choose a very hard problem:
synthesizing the whole circuit. There are no shortcuts there: The circuit do work or doesn't.

Synthetizing circuits for satisfying LTL is a very important problem. Moreover, LTL is being used for safe RL, for offering rewards to RL related to satisfying some conditions. I suggest citing this paper and relevant work:
Foundations for Restraining Bolts: Reinforcement Learning with LTLf/LDLf Restraining Specifications.
Giuseppe De Giacomo, Marco Favorito, Luca Iocchi and Fabio Patrizi. ICAPS 2019.
https://ojs.aaai.org/index.php/ICAPS/article/view/3549

I was very concerned that the data generating has turned the problem into a memorization problem. The rebuttal was good news. The method is creating new data in a very aggressive way. It's very hard to generalize when mixing assumptions and guarantees. Adjusting the assumption changes very easily the interpretation of the guarantees. Changing the guarantee suddenly matches or mismatch the assumptions. The corresponding circuit can change dramatically.

After reading the other reviews, I think the paper should explain better how hard are the instances generated.
In this context, I think it'd be enough to show how different the circuits look like when keeping the assumptions and guarantees fixed, and varying the others. In this setting, even editing distance might be informative. A more solid metric would be useful like comparing the circuits per see. Once that is done, I suggest to invite the readers to consider the risk when generating datasets for reasoning tasks.


**Main Review:**

[page 2]:* When using a beam search, models achieve an accuracy of up to

- Accuracy of correctness or recovering the label? Top accuracy? The mention of beam search might hint is top-k

-----

[page 2]:* Furthermore, the models can solve generated test instances on which classical LTL synthesis tools timed ou

- We’re the solutions correct?

-----

[page 5]:* Following the described method, we constructed a dataset containing 250000 samples split into 200000 training samples, 25000 validation samples, and 25000 test samples

- Most likely all assumptions and guarantees will occur in the validation and test sets.

-----

[page 7]:* when using a beam search algorithm only one of the predictions needs to match

- I think the statements about beam search introduce confusion. In general, using beam size k can return up to k solutions. After that, the accuracy could be measure for top-j, where j <= k. In this case, given designs can be verified afterwards, it makes sense to test all of them.
- Please, change the wording to emphasize that when using beam search, an answer is considered correct when any of the outputs is correct. That’s not the same as saying “only one of the predictions needs to match”. It’s not needed, is a decision.

-----

[page 7]: * Experiments section

- In table 1, it’s not clear that the accuracy of the last three datasets is semantic. I suggest you use different terms. For instance, “match” for syntactic accuracy, and “correct” for semantic accuracy. I think the term “total accuracy” used below is not convenient.

-----

[page 8]:* We consistently observed in all experiments that the beam search signiﬁcantly increases the accuracy.

- It’s been observed that decoding with language models with a larger beam size can decrease the quality, so this is not obvious.



**Time Spent Reviewing:**

4h

---

> ### Author Response · Authors · 2021-08-09
> **Reply to Reviewer jHAd**
>
> We thank the reviewer for their review and their helpful observations.
>
> Some clarification:
>
> - “That means that almost all assumptions and guarantees will be shared between the training and the test set.”
>
> The data generation method ensures that the combinations of assumptions and guarantees derived from the patterns are unique. The training and test set are completely disjoint. Using specification patterns is natural: a small set of specification patterns are enough to write down practical specs [12,13,23,38] (around 2000 patterns are enough to specify almost all SYNTCOMP benchmarks). We will clarify this in the paper.
>
> Main questions:
>
> - “What if Strix is given, let's say, two hours for attempting to solve the example of Fig 9, of the supplementary material?”:
>
> Please note that it is not the goal of this work to beat Strix, which has been the winner of the SYNTCOMP for years (https://strix.model.in.tum.de/). Strix can solve most of the benchmarks if given enough time (2 hours, 98% in 2020, 89% in 2021, http://www.syntcomp.org/). Figure 9 and the Timeout data set is a demonstration that the model can synthesize nontrivial circuits; and may thus prove useful for further tool development. We will clarify this more in the introduction and the experimental result section.
>
> - “Did you test in any instance out of this restriction? For instance, what's the accuracy of the rest of the SYNTCOMP formulas? Please provide separated numbers. I find the absence of this result a bit surprising. If the results were good, please add them to the paper. Otherwise, add this as a limitation of the method.”:
>
> Following [22], we capped the numbers of inputs and outputs to 5 each. Since we have to predefine the tokens for the sequence to sequence model, we cannot test on specifications with more than 5 inputs or outputs. We will clarify this limitation in the paper. Increasing the number of inputs and outputs has a non-obvious effect on the training data: The likelihood that two specification patterns share inputs and outputs decreases, making the resulting problems simpler on average. The restriction is comparable with most SYNTCOMP specs: 242 of 346 have <= 5 inputs and 274 have <= 5 outputs, but this is - obviously - a restriction that we want to lift in the future.
>
> - “I understand this is not introducing a new architecture but using an existing one. While the code is provided, the description of the parameters is extremely brief. For instance, on page 6, what does "model dimension" mean? Why this architecture? I think these deserve a better explanation. Additional details and discussions can be added in the supplementary material. If you need more space, compact fig 2 and 6.”:
>
> We followed [22] and [31] that showed great success on reasoning and symbolic tasks by applying a Transformer. The hierarchical Transformer was a valuable extension to encode the assumptions and guarantees locally before combining. A similar technique (decomposition) is used in the classical tools. We will add this discussion to the paper.
>
>
> - Main review
>
> [page2] Yes, the solutions were correct. Thanks for the other pointers! We will incorporate the suggestions in the paper.

---

> > ### Comment · Reviewer_jHAd · 2021-08-18
> > **Thank you**
> >
> > Hi, thanks for your responses. I just updated the review with some suggestions.

---

### Official Review · Reviewer_iD7H · 2021-07-17

**Rating:** 6
**Confidence:** 4

**Summary:**

This paper approaches the problem of neural circuit synthesis. To that end, the paper presents a new synthetic dataset of LTL specification and circuits/counter-examples and then, presents evaluation results by training a hierarchical Transformer model on this dataset. The synthetic dataset is generated by mining specification patterns from a real dataset (SyntComp), then artificially combining multiple instantiated patterns. The label (the circuit/counter-example) is generated using an existing classical circuit synthesis tool. The training and the model architecture are straightforward adaptations of prior work.


**Limitations And Societal Impact:**

Yes.

**Main Review:**

Strengths:
- Circuit synthesis is a new and interesting application for deep learning approaches.
- The paper presents a method to generate synthetic data for this application (which solves the issue of lack of training data).
- An implementation using hierarchical transformers that has decent performance and that can be used as a baseline for future work in this domain.
- Evaluation is extensive (includes comparing  transformers vs hierarchical transformers), multiple different kinds of datasets, and a detailed analysis of results

Weakness:

My main concern is that the lack of clarity on how the experiments are done (I am willing to increase my score if these concerns are answered in the rebuttal)
- Are the numbers in section 4 the average of multiple (ideally 5) separately trained models?
- Was the hyper-parameter search done? In a new benchmark paper, hyper-parameter search is very important to establish a good and robust baseline.
- It will be interesting to see other variations of the model architecture (for e.g. different kinds of encodings on the inputs and outputs)

Another significant concern is the lack of a discussion on the limitations to the approach
- Is it not clear how major/minor is it to have the restriction of less than 5 input and output variables?  What motivated this restriction? The limitations of the synthesis solver or the inability of the model to learn with more variables.
- What fraction of the original SyntComp dataset satisfies the filtering conditions imposed in this paper?
- Ultimately the data generation process is going to be restricted by what the synthesizer is able to solve. Although there are some cases where the learned model is able to out-beat the synthesizer, it is unlikely that it will outdo on out-of-distribution (significantly longer) specifications. Do the authors have any comments on this and how they might try to address this challenge?

Minor:
- Explain the meaning of latches
- How do the results in table 2 look for the other datasets?


**Time Spent Reviewing:**

2

---

> ### Author Response · Authors · 2021-08-09
> **Reply to Reviewer iD7H**
>
> We thank the reviewer for their review and the helpful suggestions to improve the submission!
>
> Questions:
>
> - “Are the numbers in section 4 the average[...]? Was the hyper-parameter search done?[...]”:
>
> The numbers in Section 4 are the result of a single training run. Meanwhile we have rerun the experiments as you suggested. The numbers did not change significantly, and we will update the paper accordingly.
>
> - “Is it not clear how major/minor is it to have the restriction of less than 5 input and output variables?[...]”:
>
> Increasing the number of inputs and outputs has a non-obvious effect on the training data: The likelihood that two specification patterns share inputs and outputs decreases, making the resulting problems simpler on average. So we followed [22]. The restriction is comparable with most SYNTCOMP specs: 242 of 346 have <= 5 inputs and 274 have <= 5 outputs, but this is - obviously - a restriction that we want to lift in the future. We will clarify this in the paper.
>
> - “What fraction of the original SyntComp dataset satisfies the filtering conditions imposed in this paper?”:
>
> Overall, 145 out of 346 formulas remain, so ~42% (line 292). We will make this more prominent and add how the different filters contribute to this number.
>
>
> - “Ultimately the data generation process is going to be restricted by what the synthesizer can solve. Although there are some cases where the learned model could beat the synthesizer, it is unlikely that it will outperform out-of-distribution (significantly longer) specifications. Do the authors have any comments on this and how they might try to address this challenge?”:
>
> This leads directly to our plans for future work: One (of many) directions is to incorporate the model in a training loop by generating counter example traces [22] and fast verification in between (PSPACE vs 2-EXPTIME). Proper pre-training on solid data, however, will play a crucial role for many directions, for which this paper lays the foundation.
>
> - Explain the meaning of latches
>
> A latch (or flip-flop) is a small circuit that is used as a data storage element for a single bit. The latch can thus either be in a 0 or 1 state.
>
>
> - How do the results in table 2 look for the other datasets?
>
> Specifications in the other data sets are all realizable. We will clarify this in the paper.

---

> > ### Comment · Reviewer_iD7H · 2021-08-12
> > **Response to the rebuttal**
> >
> > Thank you for your response. Here are some follow-up questions:
> >
> > 1. It is still not clear to me if the hyper-parameter search is done? If so, what is the space of the hyper-parameters?
> >
> > 2. For the experiment with multiple runs and averaging the results, is it possible for you to provide some results in the tabular format here along with standard deviations?
> >
> > 3. From your response to the limitation on the number of inputs/outputs, my understanding is that with more variables, the training distribution generated by your process is totally different from the distribution of the actual problems, which makes this data generation process useless. If that is the case, then this is a very significant limitation of your approach.
> >
> > 4. I don't understand your response to the question about being restricted by what the synthesizer can do. I don't think the current response is intended for this answer. If not, can you please elaborate more?

---

> > > ### Author Response · Authors · 2021-08-17
> > > **Reply**
> > >
> > > 1. The answer to that question was lost while copy&pasting. Sorry! We chose the parameters in accordance with related work [22]. Currently, we are implementing a hyperparameter search into our framework. We will provide the results as soon as possible.
> > >
> > > 2. Corresponding to Table 1 the following table shows the results averaged over 5 runs for beam sizes 1, 4, 8, and 16 with standard deviation in parentheses. Most results are comparable with the results reported in Table 1 except for a higher accuracy on Timeouts.
> > >
> > > Dataset | Beam Size 1 | Beam Size 4 | Beam Size 8 | Beam Size 16
> > > --- | --- | --- | --- | ---
> > > Testset | 54.65 (0.64) | 70.32 (0.63) | 74.97 (0.63) | 78.88 (0.48)
> > > SYNTCOMP | 52.69 (1.28) | 60.00 (1.45) | 65.24 (1.12) | 66.62 (0.55)
> > > Timeouts | 12.06 (1.10) | 22.28 (1.49) | 26.81 (1.78) | 31.29 (2.40)
> > > Smart Home | 18.10 (3.56) | 38.10 (6.73) | 36.19 (6.46) | 44.76 (6.46)
> > >
> > > 3. We seem to have caused a misunderstanding. We described that with more input/output variables the interaction between any two patterns becomes weaker - but this does not mean that the formulas get easier (sorry for the confusion!). As we describe in Subsection 3.2 (line 182), we gradually add patterns to generate formulas on the boundary of realizability. With a larger number of input/output variables, this process would still generate interesting formulas.
> > >
> > > 4. We apologize that our answer was not clear. One way to address the dependency on the synthesizer is to rely on (far more scalable) verification tools instead of synthesis tools, through reinforcement learning. We can generate many candidate circuits with a neural network and filter out the correct answers with the verification tool. We can then train the next neural network on those results. Even if the model only sometimes generalizes to larger/more difficult problems this process will gradually enhance its performance. Our data generation process based on specification patterns might also help this approach.

---

> > > > ### Author Response · Authors · 2021-08-25
> > > > **Hyper-parameter Search**
> > > >
> > > > The following table shows preliminary results of the hyper-parameter search we are running for the parameters embedding dimension $d_m$, feed-forward network dimension $d_{ff}$, number of local encoder layers $n_{loc}$, number of global encoder layers $n_{glob}$, number of decoder layers $n_{dec}$, and number of attention heads $n_{heads}$. Empty cells have the same value as the base model (first row). For each choice we report the accuracy on the test data for beam size 1 and beam size 16 with syntactic accuracy in parenthesis. The results are fairly stable with respect to different parameter choices. We have not yet averaged over multiple runs.
> > > >
> > > > | $d_m$ | $d_{ff}$ | $n_{loc}$ | $n_{glob}$ | $n_{dec}$ | $n_{heads}$ | Beam Size 1   | Beam Size 16  |
> > > > | ----- | -------- | --------- | ---------- | --------- | ----------- | ------------- | ------------- |
> > > > | 256   | 1024     | 4         | 4          | 8         | 4           | 51.56 (28.90) | 81.25 (39.84) |
> > > > | 128   | 512      |           |            |           |             | 50.69 (28.40)  | 76.63 (40.63) |
> > > > | 128   | 512      | 2         | 2          | 4         |             | 50.25 (28.03) | 76.60 (42.74) |
> > > > | 256   | 256      |           |            |           |             | 54.48 (30.64) | 81.48 (43.84) |
> > > > | 256   | 512      |           |            |           |             | 53.44 (30.94) | 78.59 (44.49) |
> > > > | 512   | 512      |           |            |           |             | 23.26 (4.90)  | 57.41 (26.49) |
> > > > |       |          | 2         | 2          | 4         |             | 52.80 (30.91) | 79.00 (43.08) |
> > > > |       |          | 2         | 2          |           |             | 50.59 (27.86) | 77.08 (40.51) |
> > > > |       |          | 2         | 6          |           |             | 49.85 (25.36) | 79.06 (41.19) |
> > > > |       |          | 3         | 3          | 6         |             | 50.54 (28.90) | 76.79 (40.01) |
> > > > |       |          |           |            | 4         |             | 53.75 (30.37) | 78.01 (42.01) |
> > > > |       |          | 5         | 5          | 10        |             | 15.76 (4.55)  | 45.89 (18.43) |
> > > > |       |          | 6         | 2          |           |             | 46.18 (27.25) | 74.13 (41.03) |
> > > > |       |          |           |            |           | 8           | 55.32 (31.54) | 78.91 (44.97) |
> > > > |       |          |           |            |           | 16          | 53.55 (30.27) | 78.01 (44.47) |

---

> > > > > ### Comment · Reviewer_iD7H · 2021-08-25
> > > > > **Reply**
> > > > >
> > > > > Thanks for adding the new results. They certainly increase my confidence in the results.
> > > > >
> > > > > Regarding more input/output variables --- I feel the paper will be stronger with a thorough discussion on this and if possible even repeating the experiments with a larger limit (say 8 or 10). It is fine even if the approach does not currently work with larger limits, but the results would provide great insight for the readers who want to extend your approach.
> > > > >
> > > > > Relying on better verification tools rather than better synthesizers is an interesting perspective. Although whether it actually works in practice remains to be seen.

---

### Decision · Program_Chairs · 2021-09-27

**Decision:**

Accept (Poster)

**Comment:**

This paper elicited significant discussion. On the one hand, circuit synthesis is a significant and interesting application for DL, and the technical approach is reasonable. On the other hand, there were some concerns about the level of rigor in the experiments (see the reviews for more details), and scalability of the approach. At the end, the additional clarifications/results that the authors provided during the author feedback period persuaded the reviewers and me. I am therefore recommending acceptance. Please make sure to incorporate the new results presented in your author response into the final paper.